# Diversity of the MHC class II *DRB* gene in the wolverine (Carnivora: Mustelidae: *Gulo gulo*) in Finland

Yuri Sugiyama[1], Yoshinori Nishita[2], Gerhardus M. J. Lansink[3], Katja Holmala[4], Jouni Aspi[3], Ryuichi Masuda[2]*

**1** Department of Natural History Sciences, Graduate School of Science, Hokkaido University, Sapporo, Japan, **2** Department of Biological Sciences, Faculty of Science, Hokkaido University, Sapporo, Japan, **3** Department of Ecology and Genetics, University of Oulu, Oulu, Finland, **4** Natural Resources Institute Finland (Luke), Helsinki, Finland

* masudary@sci.hokudai.ac.jp

**Data Availability Statement:** All nucleotide sequences are available from the DNA Data Bank of Japan (DDBJ) (accession numbers LC631158-LC631171).

## Abstract

The wolverine (*Gulo gulo*) in Finland has undergone significant population declines in the past. Since major histocompatibility complex (MHC) genes encode proteins involved in pathogen recognition, the diversity of these genes provides insights into the immunological fitness of regional populations. We sequenced 862 amplicons (242 bp) of MHC class II *DRB* exon 2 from 32 Finnish wolverines and identified 11 functional alleles and three pseudo-genes. A molecular phylogenetic analysis indicated trans-species polymorphism, and PAML and MEME analyses indicated positive selection, suggesting that the Finnish wolverine *DRB* genes have evolved under balancing and positive selection. In contrast to *DRB* gene analyses in other species, allele frequencies in the Finnish wolverines clearly indicated the existence of two regional subpopulations, congruent with previous studies based on neutral genetic markers. In the Finnish wolverine, rapid population declines in the past have promoted genetic drift, resulting in a lower genetic diversity of *DRB* loci, including fewer alleles and positively selected sites, than other mustelid species analyzed previously. Our data suggest that the MHC region in the Finnish wolverine population was likely affected by a recent bottleneck.

## Introduction

Wild animals, particularly large carnivores, often cause damage to livestock through predation [1], and problematic animals are harassed or killed [2], resulting in reductions in population size [3]. Depleted populations, especially larger mammals with slow reproductive rates, can then become vulnerable to extinction due to their reduced reproductive capacity [4, 5].

The wolverine (*Gulo gulo*), the largest species in the family Mustelidae, is widely distributed in the northern hemisphere, i.e. in Norway, Sweden, Finland, Russia, Mongolia, northern China, the USA, and Canada [6]. Its long-life span and low reproductive rate put populations at risk of extinction, especially those with high mortality due to poaching and legal culling [7–

**Funding:** This study was supported in part by a Joint Research Program grant from the Japan Arctic Research Network Center. The funder had no role in study design, data collection and analysis, decision to publish, or preparation of the manuscript.

**Competing interests:** The authors have declared that no competing interests exist.

9]. This animal varies physically with their habitat, leading to differences in traits like average cranial size among populations in northeastern Siberia, Norway, and Alaska [10]. Wolverines in the Finnish population, the focus of our study, ranges from 69 to 83 cm in body length (excluding the 16–25 cm bushy tail), 40 to 45 cm in height at the withers, and 8 to 28 kg in weight [11]. The wolverine is an opportunistic, facultative predator, with its diet in Finland varying from large ungulates to small rodents [12, 13]. The wolverine's distribution in northern Finland overlaps considerably with the reindeer husbandry area, and the high costs of predation in this area has led to conflict between humans and wolverines [14, 15].

The Finnish wolverine population fluctuated between 300 and 600 individuals at the end of the 1800s, began to decline in the early 1900s [16], and was reduced to 50 to 80 individuals by the 1980s [17]. In 1978, the Finnish government initiated legal protection for wolverines in all areas except the reindeer husbandry area. In 1982, with the estimated wolverine population reduced to tens of individuals, protection was extended to the whole country [18]. Wide protection led gradually to population recovery, with a marked increase since 2010 [19].

The IUCN lists for the wolverine in Europe as 'Vulnerable' [20], whereas in Finland, it is in the 'Endangered' category [21]. Currently comprising around 385 to 390 individuals, the population in Finland comprises two geographical subpopulations: 135 to 140 individuals in the reindeer husbandry area (northern subpopulation) and around 250 individuals in the area south of the reindeer husbandry area (eastern subpopulation) [22]. Analyses of microsatellite and mtDNA markers [19] have shown these two subpopulations to be genetically distinct.

Population-genetic studies on wolverines in northern Europe have utilized a variety of neutral genetic markers, including microsatellites, mtDNA, and single nucleotide polymorphisms [19, 23–25]. Finland has proven to be critical for the conservation of wolverine genetic diversity in northern Europe, as it forms a bridge between Russian and Scandinavian populations [2, 19]. Some studies have detected signs of a recent population bottleneck, reflected in an overall low genetic diversity [19, 26].

One aspect of population-genetic diversity relevant to the future viability of the wolverine in Europe is the degree of polymorphism in the major histocompatibility complex (MHC), which is related to immune fitness [27]. The MHC, the most polymorphic region in the vertebrate genome, evolved under positive and balancing selection, as shown by trans-species polymorphism (TSP) evident in phylogenetic analyses [28–30]. TSP results when alleles in an ancestral species are retained by selection in descendant species, leading to high genetic diversity in multiple genes, such as in the MHC [31].

The MHC genes, which encode proteins essential for pathogen recognition, comprise two major subfamilies, class I and class II. Class I genes mainly encode proteins presenting cytoplasmic antigens to cytotoxic T cells, whereas class II genes mainly encode proteins that present extracellular antigens to helper and regulatory T cells, thereby promoting antibody production by B cells. The functional class II protein is a heterodimer consisting of an α chain and a β chain, in the case of the DR subclass, encoded by *DRA* and *DRB* genes, respectively [32]. The antigen binding groove is formed by the α1 domain of the α chain and the β1 domain in the β chain. An amino acid residue that directly binds to an antigen is called an antigen binding site (ABS). In vertebrates, MHC polymorphism occurs predominantly at ABSs [33].

The loss of genetic diversity in a population increases its risk of extinction [34, 35]. For example, the Tasmanian devil (*Sarcophilus harrisii*) has a very low population size and low genetic diversity due to island and bottleneck effects. In this species, diversity is low in loci important for immune defense and self-recognition, such as the MHC and Toll-like receptor genes, and this is thought to contribute to the spread of diseases such as transmissible cancers [36]. MHC gene diversity is thus an important indicator in wildlife conservation research [27,

37]. The goal of our study was to investigate the genetic diversity and molecular evolution of MHC class II *DRB*s in Finnish wolverines, including trans-species polymorphism, by focusing on exon 2, which encodes ABSs.

## Materials and methods

### DNA samples

This study utilized DNA previously extracted from tissue samples from 32 individuals (Fig 1) that Lansink et al. [19] used to investigate wolverine genetic diversity and population structure in Finland. These individuals had either been hunted legally through derogation licenses, or had died in accidents and were delivered to the Natural Resource Institute Finland (Luke) research station. All individuals were obtained between 2012 and 2018, except for one from 2008 and one from 1983. The samples were assigned to "North" ($N = 11$), "East" ($N = 19$), and "Admixed" ($N = 2$) groups based on scores ($q > 0.8$) for microsatellite data previously determined by using Structure [38] at $K = 2$ (for additional details, see [19]).

### Molecular analyses

Part of MHC class II *DRB* exon 2 (242 bp, excluding the primer sequences) was amplified by PCR using forward primer Meme-DRBex2F (`CGTCCCCACAGGACATTTC`; [39]) and a reverse primer Mulu-DRBex2R (`CTCGCCGCTGCACCGTGAAG`; [40]). Reactions were performed in 25 μL volumes, each containing 5 μL of 5× PrimeSTAR GXL buffer (Takara Bio), 2 μL of dNTP mixture (2.5 mM each), 0.3 μL of phosphorylated forward and reverse primers (25 μM), 0.5 μL (1.25 U/μL) of PrimeSTAR GXL (Takara Bio), and 1–2 μL of DNA extract as the template. Thermal conditions in a Takara Thermal Cycler Dice® Touch were 94 ˚C for 2 min; 32 cycles of 98 ˚C for 10 s and 60 ˚C for 15 s; and 68 ˚C for 30 s. PCR products were confirmed by electrophoresis on 3% agarose gels and visualization with ethidium bromide under ultraviolet illumination.

For detection of *DRB* alleles, which were expected to represent multiple loci, PCR products were purified with a QIAquick PCR Purification Kit (Qiagen), ligated into pBluescript II SK + plasmid vector (Agilent Technologies), and transformed into *Escherichia coli* (JM 109) competent cells. To select cells with plasmids, bacteria were grown on 2xYT agar plates containing 50 μM ampicillin. Bacteria containing plasmids with the target PCR product were identified by blue/white selection and direct-colony PCR amplification using M13 forward and reverse primers [41]. White colonies with plasmids containing an insert were cultured in 2xYT liquid media, and plasmids were isolated with a NucleoSpin® Plasmid EasyPure column (Macherey-Nagel). Inserts were sequenced in both directions with M13 forward and reverse primers by using a BigDye Terminator v 3.1 Cycle Sequencing Kit (Thermo Fisher Scientific) and an ABI 3730 automated DNA sequencer (Applied Biosystems).

### Data analyses

Nucleotide sequences obtained were aligned by using MEGA 7 [42]. Final insert sequences were identified as potentially genuine *DRB* exon 2 sequences if they matched in the forward and reverse directions, and were detected at least twice (twice in one individual or once each from at least two individuals). Single, unique sequences were excluded, as they may have been PCR chimeras or due to other PCR errors. We confirmed candidate sequences with BLAST-N searches [43] against the National Center for Biotechnology Information (NCBI) GenBank database. Sequences verified as functional wolverine *DRB* genes or pseudogenes were named according to the MHC gene nomenclature rules for non-human species [44]. Sequences

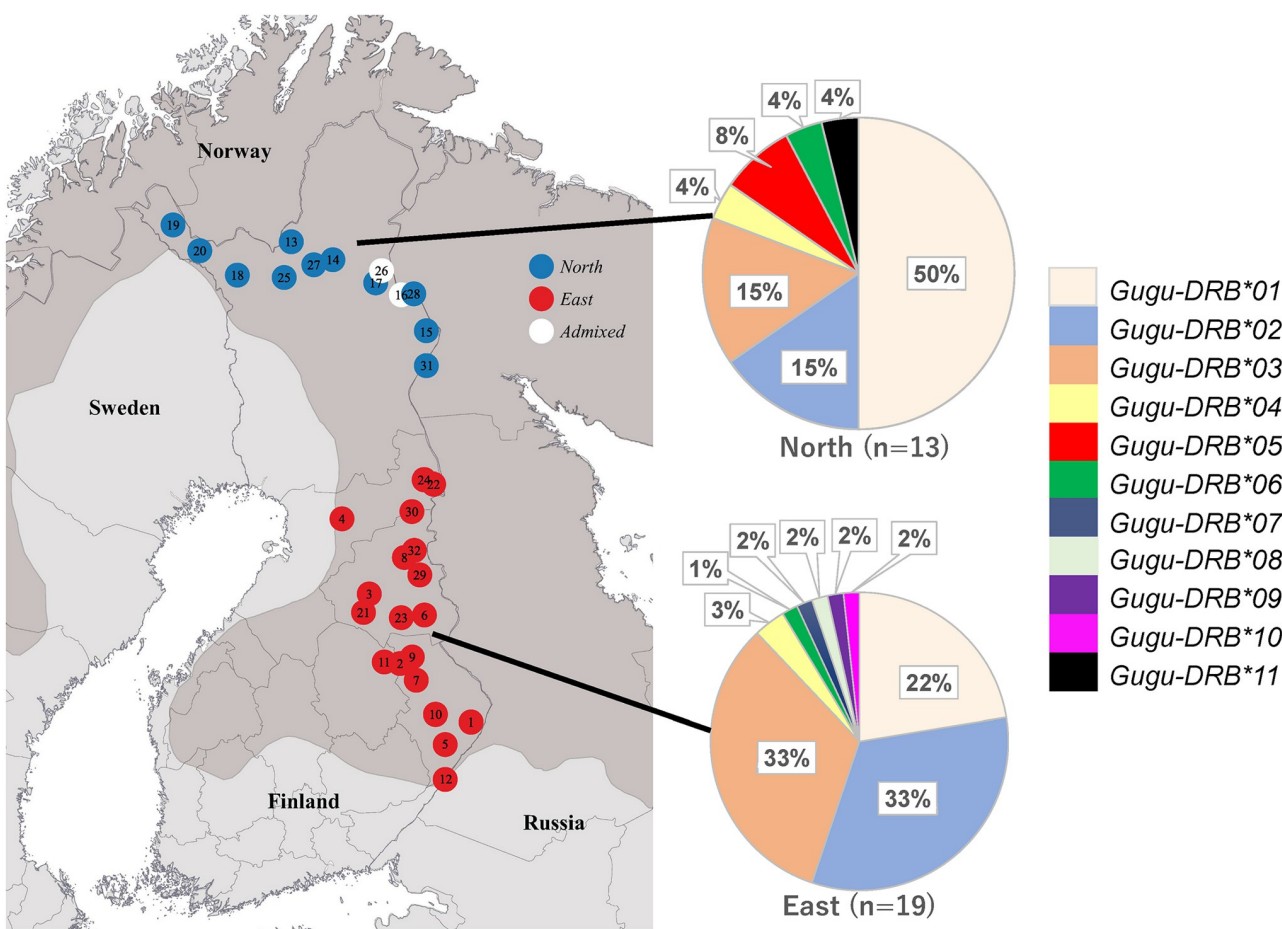

**Fig 1. Sampling locations for Finnish wolverines in this study.** Each small circle on the map indicates one individual. Based on microsatellite data [19], individuals were assigned to regional subpopulations in the north (blue, *N* = 11) or east (red, *N* = 19), or were considered to be admixed (white, *N* = 2). Inside each small circle is the sample number. The dark gray area indicates the approximate range of wolverines in Fennoscandia [19]. The pie charts show allele frequencies (in percent) in the two subpopulations, with a different color (key at the right) for each allele; admixed individuals are included in the northern subpopulation.

comprising an open reading frame (ORF) were considered to be functional genes; those with indels and/or a premature stop codon(s) were considered to be pseudogenes.

To determine whether the frequencies *DRB* alleles differed between the northern and eastern regional subpopulations, Fisher's exact test implemented in R 4.0.2 [45] was used. To check whether the mean number of alleles differed between populations, two-sample t-tests assuming unequal variances were performed with Microsoft Excel. Two individuals in the admixed group were included in the northern subpopulation due to their geographic location (Fig 1).

As an indication of selective pressure on protein-coding genes or partial sequences, the ratio ($\omega$) of nonsynonymous ($d_N$) to synonymous ($d_S$) substitution rates was calculated for ABSs, non-ABSs, and all sites. HyPhy [46] implemented in MEGA was used to detect signs of positive selection. ABSs in the wolverine MHC class II DR β-chain were inferred from those in human HLA, originally determined by a 3D structural analysis [47]. CodeML in PAML 4.8 [48] implemented in PAMLX [49] was used to investigate positive selection among all sites based on maximum likelihood methods. Four models were tested (M1a, nearly neutral; M2a,

positive selection; M7, beta; and M8, beta and ω) with likelihood ratio tests (LRTs) to determine which model best fitted our data [50–52]. Two nested models (M1a vs. M2a; M7 vs. M8) were compared by using LRTs. Positively selected sites were identified by using Bayes Empirical Bayes inference (BEB) [53]. Finally, to detect codons under positive selection, a mixed-effects model of evolution (MEME) [54] analysis was performed with Datamonkey 2.0, a web-based server for the HyPhy Package [55].

A Bayesian phylogenetic analysis was performed to reconstruct the relationships among *DRB* alleles from wolverines. The data set analyzed included alleles from Canadian and eastern Russian wolverines [56, 57], other mustelid species, and canid and felid species. Allele sequences other than those we determined were obtained from the GenBank database. To simplify the phylogenetic tree, identical sequences, regardless of species, were grouped together as a single terminal taxon. Gblocks [58] was used to eliminate inadequately aligned positions. Optimal substitution models were selected by using Kakusan4 [59] under Bayesian information criterion 4 (sample size = number of sites). The models selected were HKY85_Gamma (first and third codon positions) and K80_Gamma (second position). The phylogenetic analysis was performed by using MrBayes v3.2.7 [60] with default settings (sampling every 500 generations, with 25% burn-in), as our dataset was sufficiently large [61]. To gauge completeness of the analysis, trace files generated by the MCMC run were analyzed with Tracer v 1.7.1 [62], which determined the run to be sufficient at 80 million generations.

## Results

### Diversity of MHC class II *DRB* alleles

In all, 862 plasmid clones were sequenced from the 32 Finnish wolverines (average, 26.9 plasmid clones/individual). Eleven *DRB* exon 2 (242 bp) sequences with an open reading frame (ORF) were detected that represented functional alleles (*Gugu-DRB*01–11*) (Table 1). Another three 241 bp sequences represented presumed pseudogenes (*Gugu-DRB*PS01-03*), as all three contained a one-base deletion (Table 1) leading to a frame shift and premature stop codon(s). Five functional alleles (*Gugu-DRB*07–*11*) and one pseudogene (*Gugu-DRB*PS2*) were detected from only one individual each. All the sequences we obtained were deposited in the DNA Data Bank of Japan (DDBJ) under accession numbers LC631158–LC631171.

BLAST searches showed that five functional alleles (*Gugu-DRB*01–*05*) and one presumed pseudogene (*Gugu-DRB*PS01*) from the Finnish wolverine population were identical in 185 bp of overlapping sequence to *DRB* alleles found in Canadian and eastern Russian populations (Table 2) (the nucleotide sequences differed in length: 242 bp in our study; 185 bp for Canadian/Russian wolverines [56, 57]). Our alleles *Gugu-DRB*08* and *11* and presumed pseudogene *Gugu-DRB*PS01* were identical to those from other mustelid species (Table 2). The maximum number of alleles we found in a single individual was five, indicating at least three functional *DRB* loci per haploid genome in the wolverine. The most frequent functional allele was *Gugu-DRB*01*, followed by *02* and *03* (Fig 1 and Table 1). These three alleles were also high in frequency in the Canadian and eastern Russian populations. Because *Gugu-DRB*02* and *03* always occurred together in the same individuals, they may represent two linked loci. All 32 wolverines we examined in the Finnish population contained presumed pseudogene *Gugu-DRB*PS01*, which was also the case in the Canadian population (reported as *Gugu-DRB3*03* in [56]).

*DRB* allele frequencies differed between the two regional subpopulations in Finland (Fig 1). *Gugu-DRB*01* was detected in all individuals from the northern subpopulation, but in 13 of 19 (68.4%) individuals from the eastern subpopulation (Table 1). By contrast, both *Gugu-DRB*02* and *Gugu-DRB*03* were detected in all individuals in the eastern subpopulation, but only 4 of

**Table 1. Distribution of MHC class II *DRB* alleles and pseudogenes (*PS1*, *PS2*, and *PS3*) among 32 *Gulo gulo* individuals in Finland.**

| Subpopulation | Sample ID | Gugu-DRB* | | | | | | | | | | | | | | Total (pseudogene) |
| --- | --- | --- | --- | --- | --- | --- | --- | --- | --- | --- | --- | --- | --- | --- | --- | --- |
| | | *01* | *02* | *03* | *04* | *05* | *06* | *07* | *08* | *09* | *10* | *11* | PS1 | PS2 | PS3 | |
| Eastern | 1 | | + | + | | | + | | + | + | | | + | | | 5 (1) |
| | 2 | + | + | + | | | | + | | | | | + | | | 4 (1) |
| | 3 | + | + | + | | | | | | | | | + | | | 3 (1) |
| | 4 | + | + | + | | | | | | | | | + | | + | 3 (2) |
| | 5 | + | + | + | + | | | | | | | | + | | | 4 (1) |
| | 6 | + | + | + | + | | | | | | | | + | + | | 4 (2) |
| | 7 | + | + | + | | | | | | | | | + | | | 3 (1) |
| | 8 | + | + | + | | | | | | | | | + | | | 3 (1) |
| | 9 | + | + | + | | | | | | | | | + | | | 3 (1) |
| | 10 | + | + | + | | | | | | | | | + | | | 3 (1) |
| | 11 | + | + | + | | | | | | | + | | + | | + | 4 (2) |
| | 12 | | + | + | | | | | | | | | + | | | 2 (1) |
| | 21 | | + | + | | | | | | | | | + | | | 2 (1) |
| | 22 | + | + | + | | | | | | | | | + | | | 3 (1) |
| | 23 | | + | + | | | | | | | | | + | | | 2 (1) |
| | 24 | + | + | + | | | | | | | | | + | | | 3 (1) |
| | 29 | + | + | + | | | | | | | | | + | | | 3 (1) |
| | 30 | | + | + | | | | | | | | | + | | | 2 (1) |
| | 32 | | + | + | | | | | | | | | + | | | 2 (1) |
| | | | | | | | | | | | | | | average | | 3.1 (1.2) |
| Northern | 13 | + | | | | | | | | | | | + | | | 1 (1) |
| | 14 | + | | | | | | | | | | | + | | | 1 (1) |
| | 15 | + | | | + | | | | | | | | + | | | 2 (1) |
| | 16* | + | | | | | | | | | | | + | | | 1 (1) |
| | 17 | + | + | + | | | + | | | | | + | + | | | 5 (1) |
| | 18 | + | + | + | | | | | | | | | + | | | 3 (1) |
| | 19 | + | | | | | | | | | | | + | | | 1 (1) |
| | 20 | + | + | + | | | | | | | | | + | | | 3 (1) |
| | 25 | + | | | | | | | | | | | + | | | 1 (1) |
| | 26* | + | | | | | | | | | | | + | | | 1 (1) |
| | 27 | + | | | + | | | | | | | | + | | | 2 (1) |
| | 28 | + | | | + | | | | | | | | + | | | 2 (1) |
| | 31 | + | + | + | | | | | | | | | + | | | 3 (1) |
| | | | | | | | | | | | | | | average | | 2.0 (1.0) |

An asterisk shows an individual included in the "adimixed" group in [19].

13 (30.8%) individuals from the northern subpopulation. Individuals having both *Gugu-DRB*02* and *Gugu-DRB*03* were significantly more frequent in the eastern than in the northern subpopulation (Fisher's exact test, $P < 0.01$). Four alleles (*Gugu-DRB*07–*10*) were detected only in the eastern subpopulation and two (*Gugu-DRB*05* and *11*) only in the northern subpopulation. Significantly more individuals (Fisher's exact test, $P < 0.01$) contained only a single functional allele in the northern subpopulation (6 of 13, or 46.2%) than in the eastern subpopulation (0 of 19) individuals. Finally, there was a significant difference in the mean number of *DRB* functional alleles per individual between the eastern (3.1 alleles) and northern (2.0 alleles) subpopulations (t-test, $P < 0.05$).

**Table 2. MHC class II *DRB* alleles with identical nucleotide sequences among mustelid species.**

| *Gulo gulo* (Finland) | *Gulo gulo* (Canada,185 bp) | *Martes zibellina* | *Martes melampus* | *Mustela nivalis* | *Vormela peregusna* | *Meles leucurus* |
|---|---|---|---|---|---|---|
| *Gugu-DRB*01* | *Gugu-DRB1*04* | | | | | |
| *Gugu-DRB*02* | *Gugu-DRB2*01* | | | | | |
| *Gugu-DRB*03* | *Gugu-DRB1*02* | | | | | |
| *Gugu-DRB*04* | *Gugu-DRB2*10* | | | | | |
| *Gugu-DRB*05* | *Gugu-DRB2*07* | | | | | |
| *Gugu-DRB*08* | | *Mazi-DRB*05* | *Mama-DRB*04* | *Muni-DRB*03* | *Vope-DRB*05* | *Mele-DRB*19* |
| *Gugu-DRB*11* | | *Mazi-DRB*01* | *Mama-DRB*01* | *Muni-DRB*04* | *Vope-DRB*01* | |
| *Gugu-DRB*PS01* | *Gugu-DRB3*03* | | | | | |

*Gugu-DRB*PS01* is a putative pseudogene.

## Selection on *DRB*

The $d_N/d_S$ ratio ($\omega$) for ABS codons in Finnish wolverine *DRB*s was close to one. Values close to one indicate neutral evolution, whereas those less than one indicate negative selection and greater than one, positive selection. Our result thus indicates that the ABS codons were maintained by neutral evolution (Table 3). By contrast, $\omega$ for non-ABS codons was less than one, indicating that negative selection acted on these codons. However, PAML analyses under the M2a (positive selection) and M8 (beta and $\omega$) models provided evidence of positive selection (Table 4). Likelihood ratio tests (LRT) showed that the M2a and M8 models provided significantly better fits to our data than models without selection (Table 5). The M2a and M8 models identified four and five positively selected sites ($P < 0.05$; "+" in Fig 2), respectively, with all sites occurring in presumed ABS codons. Finally, the MEME analysis showed two codons to be under positive selection ($P < 0.05$; "+" in Fig 2), both of which were presumed ABS codons.

## Phylogeny of *DRB* alleles

In the Bayesian phylogenetic tree (Fig 3), mustelid *DRB*s comprise a monophyletic group at the family level, but not at the species or genus levels. *Gugu-DRB*05* from Finland is located in

**Table 3. Rates (±standard error) of non-synonymous ($d_N$) and synonymous ($d_S$) substitutions for antigen binding sites (ABS), non-ABS, and sites overall in the β1-domain of the MHC class II DR β-chain in *Gulo gulo*.**

| Position | Number of codons | $d_N$±SE | $d_S$±SE | $\omega$ ($d_N/d_S$) |
|---|---|---|---|---|
| Overall | 80 | 0.418±0.078 | 0.576±0.147 | 0.725 |
| ABS | 18 | 1.269±0.214 | 1.191±0.43 | 1.066 |
| Non-ABS | 62 | 0.171±0.044 | 0.398±0.134 | 0.430 |

**Table 4. Results for maximum likelihood models in CodeML.**

| models | ln*L* | Pramater estimates |
|---|---|---|
| M1a | -837.33 | $p0 = 0.566$, $p1 = 0.434$, $\omega0 = 0.000$, $\omega1 = 1.000$ |
| M2a | -832.16 | $p0 = 0.547$, $p1 = 0.279$, $p2 = 0.174$, $\omega0 = 0.000$, $\omega1 = 1.000$, $\omega2 = 3.771$ |
| M7 | -837.79 | $p = 0.007$, $q = 0.007$ |
| M8 | -832.20 | $p0 = 0.806$, $p1 = 0.194$, $p = 0.009$, $q = 0.019$, $\omega = 3.609$ |

lnL = log likelihood value; $\omega = d_N / d_S$ ratio; pn is the proportion of amino acids in the $\omega$n site class.; $p$ and $q$ are parameters of the beta distribution.

**Table 5. Results of likelihood ratio tests between two sets of nested models.**

| Model compared | 2ΔlnL | d.f. | Significance |
|---|---|---|---|
| M1a vs. M2a | 10.34 | 2 | $P < 0.01$ |
| M7 vs. M8 | 10.38 | 2 | $P < 0.01$ |

2ΔlnL = 2(lnLmodelA-lnLmodelB); d.f. = degree of freedom.

a small, near-basal clade with several other *Gugu* alleles and two from the sable (*Martes zibellina*); this clade forms an unresolved, near-basal polytomy involving other alleles from the sable, some from the European badger (*Meles meles*) and marbled polecat (*Vormela peregusna*), and the large clade containing most other mustelid *DRB* sequences. The other alleles from Finland were scattered among three distinct clades also containing *DRB* alleles from one to several other mustelid species. The three pseudogenes from Finland grouped in a clade containing only pseudogenes, including those from two other mustelid species. Both the occurrence of identical sequences between wolverines and other mustelid species (Table 2), as well as *Gugu-DRB*s more closely related to *DRB*s from other species than to those from wolverines (e.g., *Gugu-DRB*11* and *Mazi-DRB*14*), are manifestations of TSP.

## Discussion

### Differences in *DRB* allele frequencies between the northern and eastern regional subpopulations

Neutral genetic markers are useful for studying gene flow, migration, or dispersal [63], whereas MHC genes are adaptive genetic markers useful as indirect indicators of the immunological fitness of populations [27]. In highly habitat-adapted populations, MHC genes show high polymorphism, which is maintained by balancing selection [29]. Due to balancing selection, molecular phylogeographic studies based on MHC loci are expected to show less population differentiation than those based on neutral loci [64].

In previous researches on wolverines and other large carnivores, population-genetic structuring examined using MHC loci has been negligible compared to that detected through

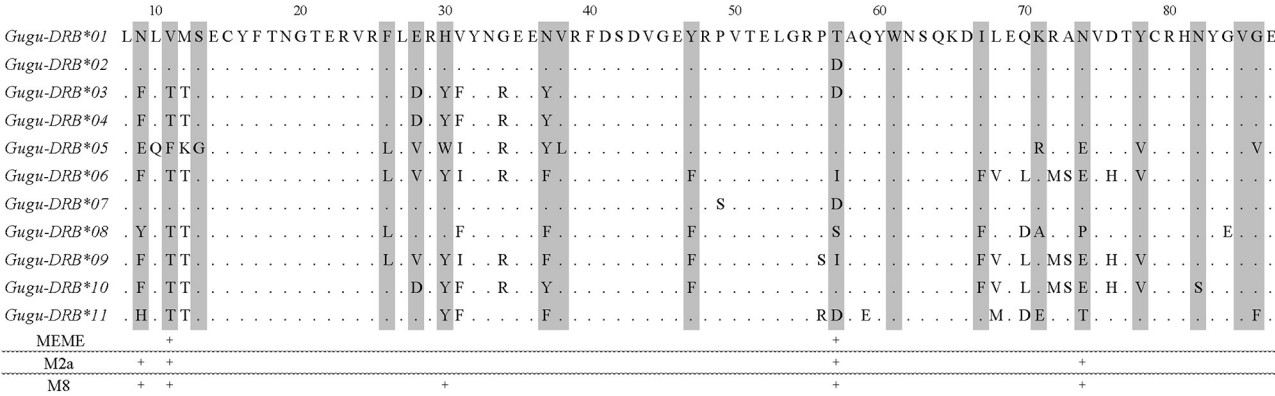

**Fig 2. Alignment of deduced amino acid sequences encoded by exon 2 for MHC class II *DRB* alleles in Finnish wolverines.** Numbers above the amino acid sequences indicate positions in the β1-domain of the DR protein β-chain. Dots indicate amino acids identical to those in *Gugu-DRB*01*. Residues in presumed antigen binding sites (ABSs) predicted from human MHC-HLA [47] are shaded in gray. + signs at the bottom of the table indicate sites inferred to be under positive selection by MEME analysis and Bayes Empirical Bayes inference (BEB) using PAML. For the M2a and M8 models in BEB, only significant results are indicated by + ($P > 95\%$).

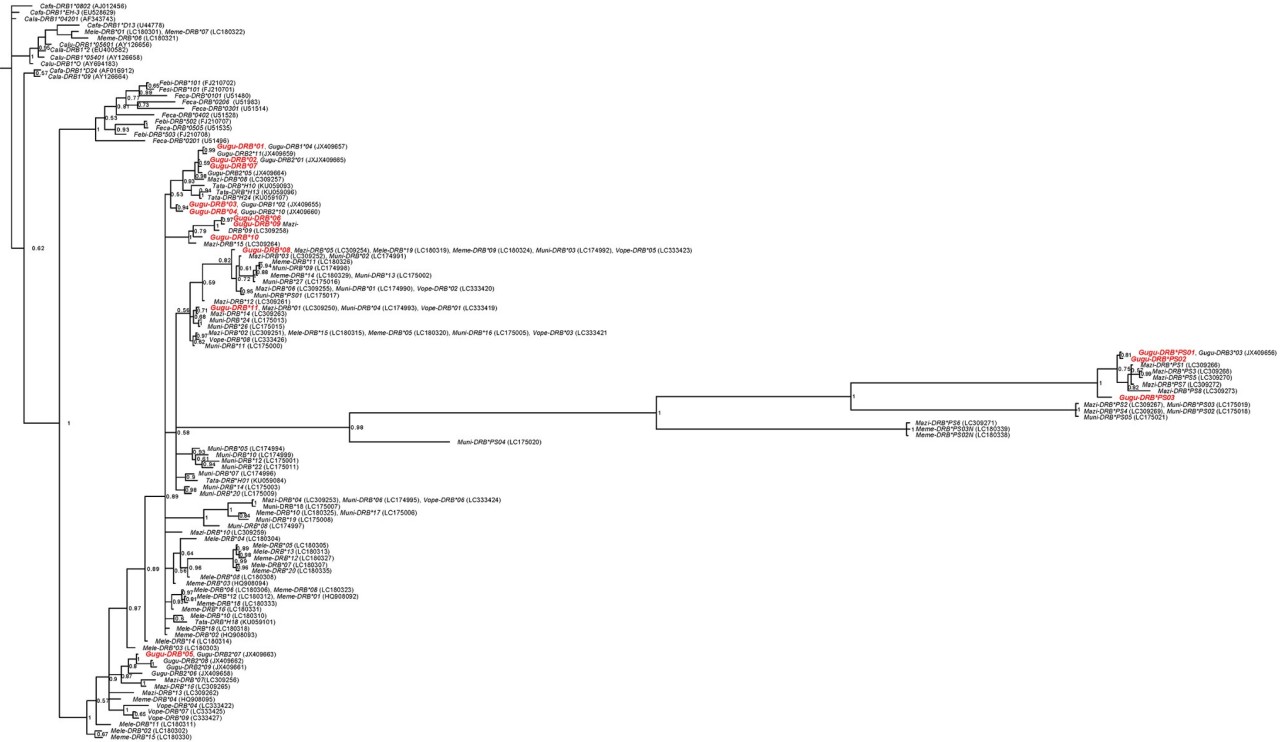

**Fig 3. Bayesian phylogenetic tree for MHC class II *DRB* exon 2 from the wolverine and other species in Mustelidae, Felidae, and Canidae.** Genbank accession numbers in parentheses follow the allele names. *Gulo gulo DRB* sequences detected in our study are in red font. Numbers near nodes are posterior probability values. The scale at the bottom indicates branch length in substitutions per site. Species abbreviations in allele names are: *Cafa, Canis familiaris*; *Cala, Canis latrans*; *Calu, Canis lupus*; *Febi, Felis silvestris bieti*; *Feca, Felis catus*; *Fesi, Felis silvestris*; *Gugu, Gulo gulo*; *Mele, Meles leucurus*; *Mazi, Martes zibellina*; *Meme, Meles meles*; *Muni, Mustela nivalis*; *Tata, Taxidea taxus*; *Vope, Vormela peregusna*.

microsatellite loci [57, 65]. Nevertheless, the distribution of MHC class II *DRB* alleles that we observed revealed genetic differences between the northern and eastern subpopulations of Finnish wolverines. These results are congruent with genetic differentiation between these subpopulations detected through microsatellites and mtDNA [19], which is believed to be linked to the severe demographic bottleneck in the 20th century. Wolverines recovered slower from the effects of the bottleneck than other sympatric species [19]. The bottleneck effect may have also contributed to genetic differentiation of the *DRB* genes between the regional subpopulations. The allele frequencies may have changed, for example, due to genetic drift after habitat loss and reduced gene flow.

## Diversity of MHC class II *DRB* alleles

We found the genetic diversity of MHC class II *DRB* loci to be lower in wolverines than in other mustelid species. In previous studies, 17 functional alleles were found among 26 Japanese martens (*Martes melampus*) [66]; 27 functional alleles among 35 least weasels (*Mustela nivalis*) [67]; and 22 functional alleles among 31 Japanese weasels (*Mustela itatsi*) [68]. By contrast, we detected only 11 functional alleles among 32 wolverines in Finland. Although different lengths of sequence were obtained, five of the 11 alleles in Finland were identical in the region of overlapping sequence to alleles in Canada [56] and eastern Russia [57]. Three of these five alleles were high in frequency in all three regions.

Using next generation sequencing, a method different from ours, Oomen et al. [56] and Rico et al. [57] detected an additional five *DRB* alleles in Canada and eastern Russia, yet across the entire region from Canada to Finland, only 16 alleles have been detected in the wolverine. Compared to other terrestrial mammals (e.g. 25 *DRB* alleles from 163 grey wolves [69], and 66 alleles from 256 raccoons [70]), including other mustelids, the wolverine shows low MHC diversity throughout its circumpolar range.

Our study showed higher genetic diversity in the eastern than in the northern subpopulation. All individuals in the eastern subpopulation had more than one functional *DRB* allele, whereas six of 13 (46.2%) individuals in the northern subpopulation carried only one functional allele, i.e. the proportion of homozygous individuals was higher in the northern subpopulation. These observations corroborate previous results from microsatellites, where the eastern subpopulation had higher genetic diversity overall than the northern [19]. In Fennoscandia, gene flow occurs between the eastern Finnish and western Russian wolverine populations via the Karelian Isthmus, whereas populations on the Scandinavian Peninsula remain fairly isolated, with no indications of gene flow—a pattern similar to that observed for European brown bears (*Ursus arctos*) [71]. Gene flow may be responsible for the higher genetic diversity in the eastern than in the northern subpopulation in Finland. Heterogeneity of MHC genes increases an individual's fitness, allowing more antigens to be recognized and increasing resistance to pathogens, e.g. *Salmonella* [72]. Our results thus suggest that the wolverines in eastern Finland may be more resistant to pathogens, especially novel or mutated ones, than their conspecifics in northern Finland. More detailed studies would be necessary to confirm this observation.

## Selection and phylogeny of *DRB* alleles

Our results were ambiguous with regard to how *DRB*s have evolved in Finnish wolverines. The *ω* value was around one at ABS codons but less than one at non-ABS codons. This could be interpreted to mean that the ABS codons evolved neutrally and non-ABS codons under purifying selection. However, the interpretation of *ω* values can be complicated. For example, a mixture of advantageous and disadvantageous mutations may also result in values less than one [51]. On the other hand, our other analyses suggested that *Gugu-DRB*s have evolved under positive selection: the PAML analysis found evidence of positive selection; the maximum likelihood codon-based selection models identified four and five positively selected sites; and the MEME analysis identified two positively selected sites. All the positively selected sites from the latter two analyses are consistent with ABSs. Nevertheless, the selection pressure has been weak, as the number of sites under selection was lower than in other mustelid species: six sites in *Martes melampus* [66] and nine in *Mustela nivalis* [67]. Finally, we should note that our study inferred ABS positions by analogy with structural studies on human ABSs, and that the exact locations of ABSs in the wolverine MHC class II DR β-chain may differ from human.

Some wolverine *DRB* alleles were identical in nucleotide sequence to those in other mustelid species, and several alleles were phylogenetically more closely related to *DRB*s of other mustelid species than to wolverine *DRB*s. This TSP among evolutionarily related species supports the conclusion that MHC genes in Finnish wolverines have been subject to balancing selection [73].

Rico et al. [57] found weak population-genetic structuring for MHC loci in Canadian wolverines and suggested that MHC variation was primarily influenced by balancing selection rather than neutral processes. However, we detected differences in genetic structure between the two regional subpopulations in Finland based on *DRB* genes. Even though population structuring is less likely to be detected at loci evolving under balancing selection [64], the

frequency differences in *DRB* loci between the Finnish regional subpopulations suggests that they might have been affected by neutral evolution. In other words, both neutral genetic drift and balancing selection may have acted simultaneously and antagonistically. Other cases of such antagonism have been reported in which genetic drift exceeded balancing selection [74, 75], or the converse [37].

In conclusion, both the study by Lansink et al. [19] using microsatellites and mtDNA to estimate genetic diversity, and our study on the diversity of MHC class II *DRB* genes, have shown that Finnish wolverine populations underwent a bottleneck that reduced diversity. Maintaining and increasing genetic diversity is important for the future conservation of wolverines in Finland.

## Acknowledgments

We thank the technical personnel at the Luke Taivalkoski Research Station for help in obtaining the DNA samples. We are grateful to Dr. Matthew Dick for giving helpful comments and editing the manuscript.

## Author Contributions

**Conceptualization:** Jouni Aspi, Ryuichi Masuda.

**Data curation:** Yuri Sugiyama, Yoshinori Nishita.

**Formal analysis:** Yuri Sugiyama, Yoshinori Nishita.

**Funding acquisition:** Ryuichi Masuda.

**Investigation:** Katja Holmala.

**Methodology:** Yuri Sugiyama, Yoshinori Nishita.

**Project administration:** Jouni Aspi, Ryuichi Masuda.

**Resources:** Gerhardus M. J. Lansink, Katja Holmala, Jouni Aspi.

**Software:** Yuri Sugiyama, Yoshinori Nishita.

**Supervision:** Ryuichi Masuda.

**Visualization:** Yuri Sugiyama, Gerhardus M. J. Lansink.

**Writing – original draft:** Yuri Sugiyama, Yoshinori Nishita, Ryuichi Masuda.

**Writing – review & editing:** Gerhardus M. J. Lansink, Katja Holmala, Jouni Aspi, Ryuichi Masuda.

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
