## [Decision Letter · Decision Letter 0]

12 Apr 2022

Diversity of the MHC class II DRB gene in the wolverine (Carnivora: Mustelidae: Gulo gulo) in Finland

PONE-D-21-33558

Dear Dr. Masuda,

We’re pleased to inform you that your manuscript has been judged scientifically suitable for publication and will be formally accepted for publication once it meets all outstanding technical requirements.

Kind regards,

Vikas Sharma, Ph.D

Academic Editor

PLOS ONE

Additional Editor Comments (optional):

The manuscript of almost ok and the reviewer has also recommended it for publication. However, authors are advised to go through entire manuscript thoroughly to remove any type of possible errors. Please check line no. 115 , Page no. 6 and correct accordingly.

Reviewers' comments:

Reviewer's Responses to Questions

**Comments to the Author**

1. Is the manuscript technically sound, and do the data support the conclusions?

Reviewer #1: Yes

2. Has the statistical analysis been performed appropriately and rigorously? 

Reviewer #1: Yes

3. Have the authors made all data underlying the findings in their manuscript fully available?

Reviewer #1: Yes

4. Is the manuscript presented in an intelligible fashion and written in standard English?

Reviewer #1: Yes

5. Review Comments to the Author

Reviewer #1: The manuscript entitled “Diversity of the MHC class II DRB gene in the wolverine (Carnivora: Mustelidae:Gulo gulo) in Finland” is well written, contains significant findings. Explored that the wolverine (Gulo gulo) in Finland has undergone significant population declines in the past and major histocompatibility complex (MHC) genes encode proteins involved in pathogen recognition, the diversity of these genes provides insights into the immunological fitness of regional populations. The results are fascinating considerate on the status of the wolverine (Gulo gulo).

6. PLOS authors have the option to publish the peer review history of their article (what does this mean?). If published, this will include your full peer review and any attached files.

Reviewer #1: No

---

## [Editor Report · Acceptance letter]

2 May 2022

PONE-D-21-33558 

Diversity of the MHC class II *DRB* gene in the wolverine (Carnivora: Mustelidae: *Gulo gulo*) in Finland 

Dear Dr. Masuda:

I'm pleased to inform you that your manuscript has been deemed suitable for publication in PLOS ONE. Congratulations! Your manuscript is now with our production department. 

Kind regards, 

on behalf of

Dr. Vikas Sharma 

Academic Editor

PLOS ONE